**HESS Opinions: Linking Darcy's equation to the linear reservoir.**
Hubert H.G. Savenije
Delft University of Technology, Delft, The Netherlands
**Abstract**
In groundwater hydrology, two simple linear equations exist describing the relation
between groundwater flow and the gradient driving it: Darcy's equation and the linear
reservoir. Both equations are empirical and straightforward, but work at different
scales: Darcy's equation at the laboratory scale and the linear reservoir at the
watershed scale. Although at first sight they appear similar, it is not trivial to upscale
Darcy's equation to the watershed scale without detailed knowledge of the structure or
shape of the underlying aquifers. This paper shows that these two equations,
combined by the water balance, are indeed identical provided there is equal resistance
in space for water entering the subsurface network. This implies that groundwater
systems make use of an efficient drainage network, a mostly invisible pattern that has
evolved over geological time scales. This drainage network provides equally
distributed resistance for water to access the system, connecting the active
groundwater body to the stream, much like a leaf is organized to provide all stomata
access to moisture at equal resistance. As a result, the time scale of the linear reservoir
appears to be inversely proportional to Darcy's "conductance"; the proportionality
being the product of the porosity and the resistance to entering the drainage network.
The main question remaining is which physical law lies behind pattern formation in
groundwater systems, evolving in a way that resistance to drainage is constant in
space. But that is a fundamental question that is equally relevant for understanding the
hydraulic properties of leaf veins in plants or of blood veins in animals.
**1. Introduction**
One of the more fundamental questions in hydrology is how to explain system
behaviour manifest at catchment scale from fundamental processes observed at
laboratory scale. Although scaling issues occur in virtually all earth sciences, what
distinguishes hydrology from related disciplines, such as hydraulics and atmospheric
science, is that hydrology seeks to describe water flowing through a landscape that
has unknown or difficult-to-observe structural characteristics. Unlike in river
hydraulics or atmospheric circulation, where answers can be found in finer grid 3-D
integration of equations describing fluid mechanics, in hydrology this cannot be done
without knowing the properties of the medium through which the water flows. The
subsurface is not only heterogeneous, it is also virtually impossible to observe. We
may be able to observe its behaviour and maybe its properties, but not its exact
structure. Groundwater is not a continuous homogeneous fluid flowing between well-
defined boundaries (as in open channel hydraulics), but rather a fluid flowing through
a medium with largely unknown properties. In other words, the boundary conditions
of flow are uncertain or unknown. As a result, hydrological models need to rely on
effective, often scale-dependent, parameters, which in most cases require calibration
to allow an adequate representation of the catchment. These calibration efforts
typically lead to considerable model uncertainty and, hence, to unreliable predictions.
But fortunately, there is good news as well. The structure of the medium through
which the water flows is not random or arbitrary; it has predictable properties that

have emerged by the interaction between the fluid and the substrate. Similar structures manifest themselves in the veins of vegetation, in infiltration patterns in the soil, and in drainage networks in river basins, emerging at a wide variety of spatial and temporal scales. Patterns in vegetation or preferential infiltration in a soil can appear at relatively short, i.e. human, time scales, but surface and subsurface drainage patterns, particularly groundwater drainage patterns, evolve at geological time scales. Under the influence of strong gradients, these patterns can evolve more quickly, but even in groundwater systems with relatively small hydraulic gradients "high permeability features" appear to be present, regulating spring flow (Swanson and Bahr, 2004).

There is a debate on the physical process causing pattern formation. Most scientists agree that it has something to do with the second law of thermodynamics, but what precisely drives pattern formation, is still debated. Terms in use are: maximum entropy production, maximum power, minimum energy expenditure (e.g. Rodriguez-Iturbe et al., 1992, 2011; Kleidon et al., 2013; Zehe et al., 2013; Westhoff et al., 2016) and the "constructal law" (Bejan, 2015). However, this paper is not about the process that creates patterns, but rather on using the fact that such patterns exist in groundwater systems to explore the connection between laboratory and catchment scale.

**How to connect laboratory scale to system scale?**
Dooge (1986) was one of the first to emphasize that hydrology behaves as a complex system with some form of organisation. Hydrologists have been surprised that in very heterogeneous and complex landscapes a relatively simple empirical law, such as the linear reservoir, can manifest itself. Why is there simplicity in a highly complex and heterogeneous system such as a catchment?

The analogy with veins in leaves, or in the human body, immediately comes to mind. Watersheds and catchments look like leaves. In a leaf, due to some organising principle, the stomata, which take $CO_2$ from the air and combine it with water to produce hydrocarbons, require access to a supply network of water and access to a drainage network that transports the hydrocarbons to the plant. Such networks are similar to the arteries and veins in our body where oxygen-rich blood enters the cells, and oxygen-poor blood is returned. The property of veins and arteries is 'obviously' that all stomata in the leaf, and cells in our body, have 'equal' access to water or oxygen-rich blood and can evacuate the products and residuals, respectively. Having equal access to a source or to a drain implies experiencing the same resistance to the hydraulic gradient. If a human cell has too high a resistance to the pressure exercised by the heart, then it is likely to die off. Likewise, too low resistance could lead to cell failure/erosion. As a result, the network evolves to an optimal distribution of resistance to the hydraulic gradient.

In a similar way, drainage networks have developed on the land surface of the Earth. Images from space show a wide variety of networks, looking like fractals. Rodriguez-Iturbe and Rinaldo (2001) connected these patterns to minimum energy expenditure. Hergarten et al. (2014) used the concept of minimum energy dissipation to explain patterns in groundwater drainage. Kleidon et al. (2013), however, showed that such patterns are components of larger Earth system functioning at maximum power, whereby the drainage system indeed functions at minimum energy expenditure.

In general, we see that patterns emerge wherever a liquid flows through a medium,
provided there is sufficient gradient to build or erode such patterns. Likewise, such
patterns must be present in the substrate through which groundwater flows, although
these are generally not considered in groundwater hydrology. If such patterns were
absent, then the groundwater system would be the only natural body without patterns,
which is not very likely.
This paper is an opinion paper. The author does not provide proof of concept. It is
purely meant to open up a debate on how the linear drainage of an active groundwater
body can be connected to Darcy's law. The discussion forum of this paper contains an
active debate between the author, reviewers and commenters that provides more
background.
**2. The linear reservoir**
At catchment scale, the emergent behaviour of the groundwater system is the linear
reservoir. Figure 1 shows a hydrograph of the Ourthe Occidentale in the Ardennes,
which on a semi-log paper shows clear linear recession behaviour, overlain by short
and fast rainfall responses by rapid subsurface flow, infiltration excess overland flow,
or saturation overland flow. The faster processes are generally non-linear, but as the
catchment dries out, the fast processes die out, the recharge to the groundwater system
stops and only the groundwater depletion remains. Even during depletion, short runoff
events may superimpose the depletion process without additional recharge, in which
case the depletion continues following a straight line on semi-logarithmic paper (see
Figure 1).
This behaviour is very common in first order streams, and even in higher order
streams. In water resources management it is well know that recession curves of
stream hydrographs can be described by exponential functions, which is congruent
with the linear reservoir of groundwater depletion. It follows from the combination of
the water balance with the linear reservoir concept. During the recession period there
appears to be a disconnect between the root zone system that interacts with the
atmosphere and the groundwater that drains towards the stream network. These two
separate "water worlds" are well described by Brooks et al. (2009) and by McDonnell
(2014) and are substantiated by different isotopic signatures. As a result, we see that
during recession only the groundwater reservoir is active.

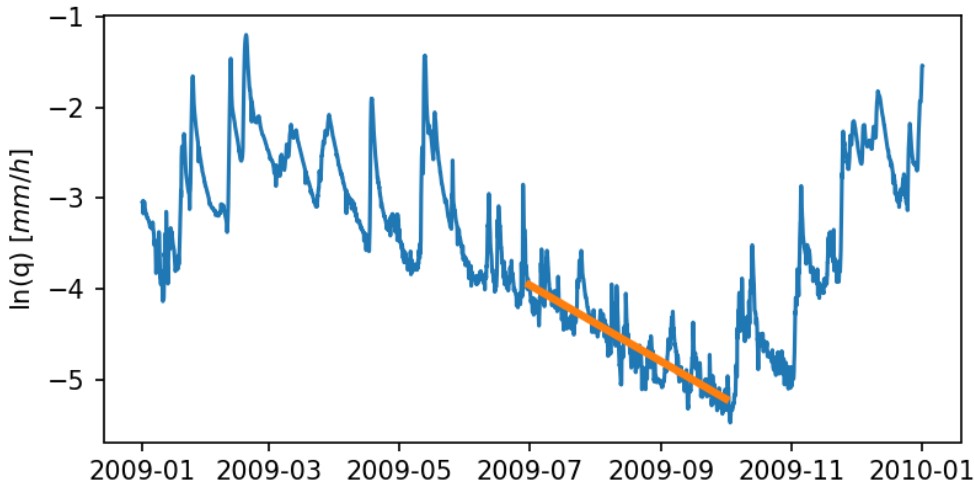

Figure 1. During the recession period, The Ourthe has a time scale of 1772 hours for groundwater depletion,
acting as a linear reservoir. Superimposed on the recession we see faster processes with much shorter time
scales.
If during recession, the catchment is only draining from the groundwater stock, then
the water balance can be described by:
$$\frac{\mathrm{d}S_g}{\mathrm{d}t} = -Q_g$$
where $S_g$ [L$^3$]is the active groundwater storage and $Q_g$ [L$^3$T$^{-1}$]is the discharge of
groundwater to the stream network.

The linear reservoir concept assumes a direct proportionality between the active (i.e.
dynamic) storage of groundwater and the groundwater flowing towards the drainage
network:
$$Q_g == \frac{S_g}{\tau}$$
where $\tau$ is the time scale of the drainage process, which is assumed to be constant.
Combination with the water balance leads to:
$$Q_g = Q_0 \exp\left(-t/\tau\right)$$
where $Q_0$ is the discharge at $t$=0. So the exponential recession, which we observe at
the outfall of natural catchments, is congruent with the linear reservoir concept. But
how does this relate to Darcy's law, which applies at laboratory scale?

**3. Upscaling Darcy's law**

Darcy's law reads:
$$\overline{v} = -k\frac{\mathrm{d}\varphi}{\mathrm{d}x}$$
where: $\overline{v}$ is the discharge per unit area, or filter velocity [LT$^{-1}$]; $k$ is the conductance
[LT$^{-1}$]; $\varphi$ is the hydraulic head [L]; and $x$ [L] is the distance along the stream line. In
a drainage network, these streamlines generally form semi-circles, perpendicular to
the lines of equipotential, draining almost vertically downward from the point of
recharge and subsequently upward when seeping to the open drain (see Figure 2 for a
conceptual sketch).

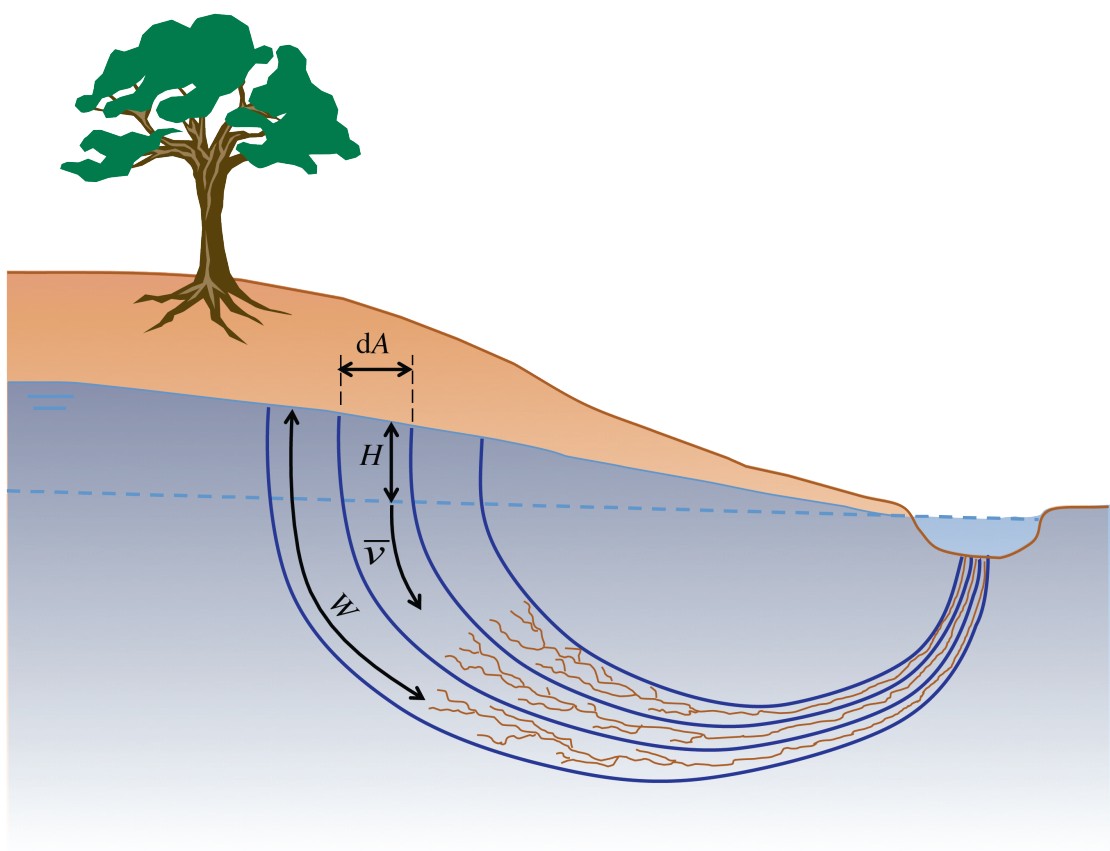

**Figure 2. Conceptual sketch of an unconfined freatic groundwater body draining towards a surface drain.**
**H is the head of the freatic water table with respect to the nearest open water.**
Henry Darcy (1803-1858) found this relationship under laboratory conditions, but the
law also appears to work fine in regions with modest slopes, where one or more layers
can be identified with conductivities representative for the sediment properties of
these layers. In such relatively flat areas, upscaling from the laboratory scale to a
region with well-defined layer structure appears to work rather well. This is clear
from the many groundwater models, such as MODFLOW, that do well at representing
hydraulic heads. However, such regional groundwater models are generally calibrated
solely on water levels (hydraulic head) and seldom on flow velocities, transport of
solutes, or flows, leading to equifinality in the determination of spatially variable $k$
values.
Swanson and Bahr (2004) identified preferential flow even in mildly sloping terrain.
Therefore it is reasonable to assume that under stronger gradients preferential flow
becomes more prominent. In sloping areas, the hypothesis is that the subsurface is
organised and cannot be assumed to consist of layers with relatively homogeneous
properties. Under the influence of a stronger hydraulic gradient, drainage patterns
occur in the substrate more or less following the hydraulic gradient along the
streamlines. This happens everywhere in nature where water flows through an
erodible or soluble material. An initial disturbance leads to the evolution of a drainage
network that facilitates the transport of water through the erodible material. Initial
disturbances can be cracks, sedimentation patterns, animal burrows, former root
channels, etc. The formation of the network can be by physical erosion and deposition
(breaking up, transporting and settling particles) but can also be by chemical activity
(minerals going into solution or precipitating). The latter is the dominant process in
groundwater flow. The precipitation that enters the groundwater system through
preferential infiltration (Brooks et al. 2009; McDonnell, 2014) is low in mineral
composition and hence aggressive to the substrate. The minerals that we find in the
stream during low flow (when the river is fed by groundwater) are the erosion
products of the drainage network being developed. In the mineral composition of the
stream we can see pattern formation at work and from the transport of chemicals by
the stream we may derive the rate at which this happens.
In contrast to the physical drainage structures that we can see on the surface (e.g. river
networks, seepage zones on beaches, etc.), sub-surface drainage structures are hard to
observe. But they are there. On hillslopes, individual preferential sub-surface flow
channels have been observed in trenches, but complete networks are hard to observe
without destroying the entire network.
The hypothesis is that under the ground a drainage system evolves that facilitates the
transport of water to the surface drainage network in the most efficient manner. As
was demonstrated by Kleidon et al. (2013) an optimal drainage network maximizes
the power of the sediment flux, which involves maximum dissipation in the part of the
catchment where erosion takes place and minimum energy expenditure in the
drainage network. This finding is in line with the findings of Rodriguez-Iturbe and
Rinaldo (1997, p.253), who found that minimum energy expenditure defines the
structure of surface drainage. Although a surface drainage network has 2-D
characteristics on a planar view, the groundwater system has a clear 3-D drainage
structure. The boundary where open water and groundwater interact also has a
complex shape. This is the boundary where the groundwater seeps out at atmospheric
pressure indicated in Figure 2 by the dotted blue line. This boundary of interaction
follows the stream network and moves up and down with the water level of the
stream. To describe this 3-D drainage network conceptually, we can build on the
analogy with a fractal-like (mostly 2-D) structure of a leaf or a river drainage
network, but it is not the same.
Fractal networks can be described by width functions that determine the average
distance of a point to the network. Let's call this distance W. Let's now picture a
cross-section over a catchment with an unconfined phreatic groundwater body
draining towards an open water drain (see Figure 2 for a conceptual sketch). At a
certain infinitesimal area d$A$ of the catchment, the drainage distance to the sub-surface
network is $W$. The head difference to the nearest open drain is $H$. Darcy's equation
then becomes:

$$\bar{v} = k\frac{H}{W} = \frac{H}{r_g}$$

where $r_g$ [T] is the resistance against drainage. This way of expressing the resistance
is similar to the aerodynamic resistance and the stomatal resistance of the Penman-
Monteith equation. It is the resistance of the flux to a difference in head. So, instead
of assuming a constant width to the drainage network, we assume a constant
resistance to flow. This is in fact the purpose of veins in systems like leaves or body
tissues, such as lungs or brains or muscles. The veins make sure that the resistance of
liquids to reach stomata in the leaf, or cells in living tissue, is optimal and equal
throughout the organ. But also in innate material, where gravity and erosive powers
have been at work for millenia, the system is evolving towards an equally distributed
resistance to drainage, much in line with the minimum expenditure theory of
Rodriguez-Iturbe and Rinaldo (1997).
Building on Darcy's equation, an infinitesimal area $dA$ of a catchment drains:
$$dQ_g = \bar{v}\, dA$$
Interestingly, this drainage (recharge to the groundwater) is downward, so that we can
assume that $dA$ lies in the horizontal plane. If we integrate the discharge over the area
of the catchment that drains on the outfall, and assuming a constant resistance, we
obtain:
$$Q_g = \int_A \bar{v}\, dA = \frac{1}{r_g} \int_A H\, dA = \frac{\bar{H}A}{r_g} = \frac{S_g}{r_g n}$$
where $n$ [-] is the average porosity of the active groundwater body (which is the
groundwater body above the drainage level). We see that the areal integral of the head
$H$ equals the volume of saturated substrate above the level of the drain. Multiplied by
the porosity, this volume equals the amount of groundwater stored above the drainage
level, which equals the active storage of groundwater $S_g$. Comparison with the linear
reservoir provides the following connection between the system time scale $\tau$, the
resistance $r_g$ and the average porosity $n$:
$$\tau = \frac{W}{k} n = r_g n$$
As a result, we have been able to connect the time scale of the linear reservoir to the
key properties of Darcy's equation, being the average porosity, the conductance and
the distance to the sub-surface drainage structure, or better, to the average porosity
and the resistance to drainage. This resistance to drainage is assumed constant in
space, but will evolve over time, as the fractal structure expands. However, at a
human time scale, this expansion may be considered to be so slow that the system can
be assumed to be static.
**4. Discussion and conclusion**
In groundwater flow, connecting the laboratory scale to the system scale requires
knowledge on the structure, shape and composition of the medium that connects the
recharge interface to the drain. Here we have assumed that, much like we see in a
homogenous medium, the flow pattern follows streamlines perpendicular to the lines
of equal head, forming semicircle-like streamlines. This implies that flow in the upper
part of the streamlines is essentially vertical and that integration of Darcy's law over
the cross-section of a stream tube takes place in the horizontal plane, and not in a
plain perpendicular to the gradient of the hillslope.
The second assumption is that, over time, patterns have evolved along these
streamlines by erosion of the substrate. It is then shown that if the resistance to flow
between the recharge interface and the drainage network is constant over the area of
drainage, that the linear reservoir equation follows from integration. This constant
resistance to the hydraulic gradient is similar to what we see in leaves or body tissue.
What is the evolutionary dynamics of the drainage network? It is likely that the
drainage network makes use of cracks and fissure present in the base rock, but
subsequently expands and develops by minerals going into solution. As a result, these
networks never stop to develop, continuously refining and expanding the fractal
structure. In relatively young catchments such structures may not be fully developed.
By sampling the chemical contents of springs and base flow at the outfall of
catchments, we may be able to determine the rate of growth of the drainage network,
and -- if the mineral content of the substrate is known -- the origin of the erosion
material. I think it is an interesting venue of research to study the expansion of such
networks as a function of the mineral composition of the groundwater feeding the
stream network, possibly supported by targeted use of unique tracers.
This paper does not provide an explanation for the fact that in recharge systems
groundwater drains as a linear reservoir. In fact, it raises more fundamental questions:
If a catchment has exponential recession, congruent with a linear reservoir, then what
causes the resistance to entering the drainage network to be constant? What is the
process of drainage pattern formation? If the sub-surface forms fractal-like structures,
then which formation process lies behind it? The reason why this property evolves
over time is still to be investigated, but it is likely that the reason should be sought, in
some way or another, in the second law of thermodynamics.
We know from common practice that in mildly sloping areas, groundwater models
that spatially integrate Darcy's equation are quite well capable of simulating
piezometric heads. We also know that predicting the transport of pollutants in such
systems is much less straightforward, requiring the assumption of dual porosities
(which are in fact patterns). In more strongly sloping areas, such numerical models
are much less efficient to describe groundwater flow. This can, of course, be blamed
on the heterogeneity of the substrate, but one could also ask oneself the question if
direct application of Darcy's law is the right approach at this scale. If under the
stronger gradient of a hillslope preferential flow patterns have developed, then we
should take the properties of these patterns into account. Fortunately, nature is kind
and helpful. It has provided us with the linear reservoir that we can use as an
alternative for a highly complex 3-D numerical model that has difficulty to reflect the
dual porosity of patterns that we cannot observe directly, but of which we can see its
simple signature: the linear reservoir with exponential recession. Hopefully
groundwater modellers are going to make use of that property, particularly in larger
scale modelling studies.

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
