# Peer review of "HESS Opinions: Linking Darcy's equation to the linear reservoir."

_Hydrology and Earth System Sciences, 2017_

## Short Comment (SC1) · 6 Oct 2017

In this short, thought-stimulating, opinion paper, Savenije reasons how two key equations in hydrology at different scales (i.e. the linear reservoir at catchment –scale and Darcy's equation at lab-scale) are connected. Understanding the connections between scales in hydrology, and the cause of emergent catchment behavior is very valuable for hydrological sciences to progress. This paper makes an interesting contribution to this challenge. In general, I really enjoyed reading this short piece (and an earlier presentation of this work at the EGU General Assembly 2017 in Vienna, catchment similarity session).

Savenije writes that this is an "opinion paper" that "does not provide a proof of concept",

and is intended to "open a debate on how the linear drainage of groundwater from a hillslope can be connected to Darcy's law". I spirit of this comment, I have a few thoughts that are intended as thought stimulating comments that may hopefully further strengthen this paper or this overall debate.

**1. Is the catchment's groundwater reservoir linear?** The paper states "At catchment scale, the emergent behaviour of the groundwater system is the linear reservoir" and makes similar assertions in other places too (e.g. line 139). This key premise is qualitatively supported by Figure 1 with data from the Ourthe. However, is this premise really representative for most (/many) places?

**From an empirical perspective**, most studies that systematically characterized the groundwater contribution to streamflow do not find linear reservoir behavior in most catchments (e.g.: Brutsaert  Nieber, 1977; Ye et al., 2014; Berghuijs et al., 2016).

**From a theoretical perspective**, we should also not expect the groundwater reservoir to behave linearly; several linear reservoirs assembled together will create a non-linear overall response. This can be shown using straightforward math and has been discussed in the context of hydrology by Harman et al. (2009), explaining how power law catchment-scale recessions arise from heterogeneous linear small-scale dynamics.

**2. Is upscaling Darcy flow a logical choice in describing subsurface drainage networks?** The paper suggests that (while difficult to observe) sub-surface drainage structures are largely preferential (e.g. "on hillslopes, individual preferential sub-surface flow channels have been observed in trenches, but complete networks are hard to observe without destroying the entire network.") To me this implies that upscaling Darcy flow is maybe not the right approach to describe flow processes, since the processes you describe the network to consist of are all preferential (instead of Darcy flow?).

**3. Are areas far away from the stream contributing more to total groundwater flow reaching the streams?** By assuming a constant resistance for the entire catchment, it implies that areas further away from the stream disproportionally contribute to

GW flow to stream? (since they will have a bigger head difference with the stream) Is this consistent with our perceptual model of catchments? and with tracer data-based studies?

**4. Is resistance constant?** Aquifer conductivities can vary by many orders of magnitude [Gleeson et al., 2011], even within an individual catchment [Ameli et al., 2016]. Is this consistent with what the assumption of constant resistance? Would this imply that aquifer conductivities are highest for the shortest flow paths? Is this something we observe in nature? It seems from Ameli that highest resistance is in deeper parts of the aquifer instead?

**5. Is groundwater recharge vertical?** The analysis assumes that GW recharge is vertical. In my interpretation, this assumption should also hold at other depths than the infiltration surface for the calculation to work? (call me out if I'm wrong here and stop reading this comment if that's the case. Otherwise, continue reading). However, is this realistic in most landscapes? In landscapes where horizontal flow path lengths are significant compared to vertical flow path lengths, this key assumption seems violated. A quick back of the envelope calculation of this seems to suggest that this assumption is violated in most landscapes? For example, when looking at Wang and Wu (2013) the average stream density for MOPEX catchments (approx. 1 km/km2?) implies that groundwater, on average, also travels kilometers deep (and up again) before contributing to streamflow? This seems unrealistic to me in most landscapes?

**6. The paper talks about "residence time", but this term may confuse part of the community**. In several parts of the paper, residence time is used to refer to the "characteristic time-scale of the linear reservoir". This is confusing because "residence time" in hydrology is commonly used to refer to the ages of water stored in a catchment (e.g. Rinaldo et al. 2011). Therefore, I recommend not to use "residence time" when you describe flow processes (instead of transport).

Overall, I really enjoyed reading the paper, and the above comments are intended

as my insignificant contribution to the "open a debate on how the linear drainage of groundwater from a hillslope can be connected to Darcy's law".

**References**

Ameli, A. A., J. J. McDonnell, and K. Bishop (2016), The exponential decline in saturated hydraulic conductivity with depth: A novel method for exploring its effect on water flow paths and transit time distribution, Hydrol. Processes, 30, 2438–2450.

Berghuijs, W. R., Hartmann, A., Woods, R. A. (2016). Streamflow sensitivity to water storage changes across Europe. Geophysical Research Letters, 43(5), 1980-1987.

Brutsaert, W., Nieber, J. L. (1977). Regionalized drought flow hydrographs from a mature glaciated plateau. Water Resources Research, 13(3), 637-643.

Gleeson, T., L. Smith, N. Moosdorf, J. Hartmann, H. H. Dürr, A. H. Manning, L. P. H. van Beek, and A. M. Jellinek (2011), Mapping permeability over the surface of the Earth, Geophys. Res. Lett., 38, L02401, doi:10.1029/2010GL045565.

Harman, C. J., Sivapalan, M., Kumar, P. (2009). Power law catchment‐scale recessions arising from heterogeneous linear small‐scale dynamics. Water Resources Research, 45(9).

Rinaldo, A., K. J. Beven, E. Bertuzzo, L. Nicotina, J. Davies, A. Fiori, D. Russo, and G. Botter (2011), Catchment travel time distributions and waterflow in soils, Water Resour. Res., 47, W07537, doi:10.1029/2011WR010478.

Ye, S., Li, H. Y., Huang, M., Ali, M., Leng, G., Leung, L. R., ... Sivapalan, M. (2014). Regionalization of subsurface stormflow parameters of hydrologic models: Derivation from regional analysis of streamflow recession curves. Journal of hydrology, 519, 670-682.

Wang, D. and Wu, L.: Similarity of climate control on base flow and perennial stream density in the Budyko framework, Hydrol. Earth Syst. Sci., 17, 315-324,

https://doi.org/10.5194/hess-17-315-2013, 2013.
* * *

---

## Short Comment (SC2) · 10 Oct 2017

I thank Prof. Savenije for so quickly providing these answers and clarifying some of my misunderstandings of this work. Given the complex nature of the addressed question, several unknowns (will always) remain to exist. However, the provided answers are overall satisfactory to me. Well done!

---

## Short Comment (SC3) · 10 Oct 2017

M. Cuthbert

cuthbertm2@cardiff.ac.uk

This is a thought provoking opinion paper by Hubert Savenije (H.S.) concerning how Darcy's law relates to the linear reservoir concept. The response by Wouter Berghuijs (W.B.) provides a useful basis for discussing a number of the assumptions made in the paper, and while I agree with most of the points raised in that response, I have a few other thoughts to add to the discussion. I've used the same numbered points used by W.B. to maintain the flow of the discussion:

1. Is the catchment's groundwater reservoir linear?

I agree with W.B. that empirically, linear reservoir responses may not often be observed and that theoretically, there are many reasons why a groundwater will not give a linear

response (e.g. see refs in W.B.'s comment). However, there are sound reasons why groundwater may sometimes behave like a linear reservoir and these have already been linked in the literature to Darcy's law via the linearised Bousinesq equation (e.g. various cases shown by Brutsaert (2005)). Most simply, where groundwater flow is predominantly horizontal (with characteristic length, L) the groundwater hydraulic response time is found to be proportional to $L^2/D$, with D being the hydraulic diffusivity (e.g. Erskine & Papaioannou (1997)). I think though that this type of linear reservoir concept and response time formulation is only valid for small and/or highly hydraulically diffuse situations (Cuthbert 2014), although it may also occur in 2-D radially convergent or divergent settings, not just for 1-D flow (Cuthbert 2014).

The result given in the opinion paper that 'K = rg.n' is thus rather different to that given in previous literature. The given formulation of the response time is still dependent on the hydraulic properties (porosity – which is implicitly assumed equal to specific yield in the paper - and hydraulic conductivity). However, here it is related to the length scale of the flow domain (vertically) as opposed to the square of the characteristic length (horizontally). This difference follows from the assumption in the paper that groundwater will drain vertically to the nearest preferential flow pathway and that the head variation varies linearly over this flow path. It appears to be mathematically equivalent to equating the catchment drainage response to a 'falling head permeameter'.

(As a minor aside, perhaps it would be better to call the response time something like 'Tau' in the paper rather than K which is normally reserved for hydraulic conductivity, to avoid confusion?)

2. Is upscaling Darcy flow a logical choice in describing subsurface drainage networks?

The key question here is how the preferential flow network relates hydraulically to the rest of the connected subsurface porosity. Does it behave like an 'equivalent porous media'? Or does it exhibit explicit features expected of a dual porosity system? Both these concepts are mature in the hydrogeological literature and should be brought into

the discussion. . .

3. Are areas far away from the stream contributing more to total groundwater flow reaching the streams?

W.B. comments here "By assuming a constant resistance for the entire catchment, it implies that areas further away from the stream disproportionally contribute to GW flow to stream? (since they will have a bigger head difference with the stream)".

I'm not sure I agree here. If the resistance is the same everywhere and the preferential flow paths have a much greater hydraulic conductivity than the rest of the subsurface, then surely the result will be that the 'water table' has almost no slope – there is no need for the heads further away from the stream to be significantly higher than the head in the stream? If there is a significant difference between the water table gradient and the piezometric profile in the preferential flow zone as shown in Figure 2, then there must also be a component of horizontal flow occurring in the 'non-preferential' zone? This rather undermines the conceptual model? I think the conceptual model and Figure 2 needs more thought.

In essence I would expect the hydraulic response at the stream to be a complex interaction between the hydraulic response in the higher and lower permeability zones depending on their relative hydraulic conductivity and the vertical and horizontal length scales in question.

An interesting study with similarities to the proposed conceptual model (i.e. highly conductive preferential flow pathways extending through the subsurface from a drainage point into the catchment) has been studied by Swanson & Bahr (2004). They found that the $L^2/D$ relationship still holds in such a case, at least for the range of parameters studied, even though the head distribution is modified by the presence of the preferential pathway.

4. Is resistance constant?

My comment on point 2 above is relevant here – whether this is reasonable depends on the scale of the heterogeneity versus the scale of the catchment observations.

5. Is groundwater recharge vertical?

The terminology in the paper is confusing here. 'Recharge' is not the same as 'drainage' or 'groundwater flow'. I think the paper is suggesting that groundwater flow is predominantly vertical through the bulk of the subsurface but predominantly horizontal through the preferential pathways - the direction of the recharge arriving through the unsaturated zone to the water table doesn't seem relevant to the argument?

6. The paper talks about "residence time", but this term may confuse part of the community

I totally agree with W.B. that this is adding unnecessary confusion. The "characteristic time-scale of the linear reservoir" is a hydraulic response time (timescale of a pressure wave propagation) which is a completely different concept to residence time (related to the velocity of water molecules).

Summary

In summary, I would suggest that there are a whole range of conceptual models (and related mathematics) which can link Darcy's law to the linear reservoir equation. The one presented in this opinion piece may be amongst them. However I would encourage thinking along the lines of there being a continuum of catchment hydraulic responses ranging from those where multi-porosity is explicitly exhibited in the groundwater discharge response (which may be most similar to the idea presented here) and those for which the discharge response looks more like an equivalent porous medium consistent with previous research on this. While we may be able to imagine end members for this continuum and find real examples in nature, I imagine most catchments will exhibit and integrate both behaviours at the same time at the typical spatial scale sampled by a stream flow gauge.

Apologies if I've misunderstood any of the points presented in the paper by H.S. or the comment by W.B., but I hope my comments are of some use in the ongoing discussion of this theme.

References

Brutsaert, W. (2005), Hydrology: An Introduction, Cambridge Univ. Press, Cambridge, U. K.

Cuthbert, M. O. (2014). Straight thinking about groundwater recession. Water Resources Research, 50(3), 2407-2424.

Erskine, A. D., & Papaioannou, A. (1997). The use of aquifer response rate in the assessment of groundwater resources. Journal of Hydrology, 202(1), 373-391.

Swanson, S. K., & Bahr, J. M. (2004). Analytical and numerical models to explain steady rates of spring flow. Groundwater, 42(5), 747-759.

---

## Author Comment (AC1) · 10 Oct 2017

Thank you very much for this constructive and critical reaction to the opinion paper. This is much appreciated. Being written as an opinion paper, it is exactly the intention of the paper to trigger reactions by the community, in the hope that we can address one of the biggest riddles in hydrology that I have been struggling with. The riddle that I tried to solve is why so many catchments demonstrate linear behaviour. Indeed, as Berghuijs indicates, there are quite a number of catchments that are not linear, but have a power of $n=2$ (corresponding with $b=1.5$ in the $dQ/dt$ versus $Q$ plots). This power of 2 is in some agreement with the Boussinesq equation for a sloped aquifer with an impermeable basement (Verhoest and Troch, 2000). However, the fact remains that many catchments demonstrate linear reservoir behaviour, and we don't know where

this comes from. Probably the reason lies in the conditions for the Boussinesq equation to apply. It depends on how permeable the underlying aquifer is, and how strong the slope is. And maybe these two characteristics are correlated. An impermeable foundation deflects the streamlines, whereby the groundwater flow at the start of the streamlines is not purely vertical.

**1. Is the catchment's groundwater reservoir linear?**

Berghuijs states that many catchments do not perform like linear reservoirs, but then on the other hand, many do. As was indicated by Ye et al. (2014), a substantial number of catchments have a power ($n$) between 0.8 and 1.3. They showed that this power depends strongly on slope (the more sloped, the less linear) and on the aridity index (higher aridity, more linear). These two indicators may be correlated, because sloped terrains in the west of the USA and in the Rocky mountains are seldom dry. A strongly sloping catchment with a poorly permeable base rock is indeed likely to function according to the Boussinesq equation (e.g. Verhoest and Troch, 2000). But catchments that are more similar to the situation sketch in the opinion paper (Figure 2) with a deep freatic aquifer and –as a result –a dominant vertical flow direction above the level of the nearest open water, appear to function as linear reservoirs.

By the way, looking more closely at the figures in Brutsaert and Nieber (1977), lines with $b$=1 would fit almost as well to Figs. 4, 5, 6 and 7 of that paper. Only Figs. 2 and 3 have a clearly identifiable steeper slope with $1<b<3/2$. But $b$=3/2 is clearly on the high side. In general the lines drawn in these figures are suggestive.

I agree with Berghuijs that adding up two exponential equations does not generally result in another exponential function. There are two conditions where it does: when the time scales of the two catchments are the same; and when one catchment is generating much more flow than the other. But this is of course not a satisfactory answer. A more interesting possibility would be that the two catchments influence each other, and that they work in tandem. In a drainage network, a fractal-like network, the groundwater is drained in a complex three-dimensional pattern, whereby it is not unthinkable that two neighbouring drainage basins interact during low flow. Streamlines are likely to bend off as the water levels in the drainage network subside. How this exactly works, is not easy to figure out, but the fact that we see linear reservoir behaviour also in composite catchments, indicates that some interaction is taking place. So, to be honest, I don't know the answer to the question, but I do know that there is an interesting riddle to be solved.

**2. Is upscaling Darcy flow a logical choice in describing subsurface drainage networks?**

Beghuijs raises the question whether Darcy's law is the right equation to use if it is assumed that groundwater flow is preferential. This is a misunderstanding. The idea is that Darcy's flow only applies for the groundwater to reach the preferential drainage network. The analogy with the blood vessel system is that recharge to the soil and the groundwater level is preferential (like the artery system) and that the drainage network to the stream is also preferential (as the veins). But in between the flow is Darcyan. The drainage network does not extend all the way to the water table, but only starts a certain distance $W$ away from the groundwater table. Over that distance the flow is Darcyan, until the point where it accesses the drainage network. The consequence of using both the linear reservoir and Darcy's equation is that the resistance to entering the network is the same all over the domain of integration.

**3. Are areas far away from the stream contributing more to total groundwater flow reaching the streams?**

This is a very interesting point. It is reasonable to assume that the drainage network far away from the open water is not as well established as the network closer to the stream. Thus the distance $W$ to entering the network is longer. The larger head is then offset against a longer travel distance for the Darcyan flow.

**4. Is resistance constant?**

But clearly closer to the stream, the travel distance is smaller. However we also know that closer to the stream the soils are less permeable, with a lower conductance. Also, as Berghuijs correctly observes, there is a lot of heterogeneity in the groundwater system. But maybe what some define as heterogeneity is in fact the manifestation of patterns, where in some parts the conductance seems very high (if there are veins) and in some places the conductance is low (relatively dead pockets). Of course, the sketch in Figure 2 is just a simple impression of how it might look. In reality the system of preferential drainage will be difficult to map and in reality may look capricious. The essential assumption here is that if the linear reservoir applies to systems without an impermeable base rock, then the resistance to entering the preferential drainage network should be constant.

**5. Is groundwater recharge vertical?**

The assumption is that between the freatic water table and the zero head level of the nearest drainage (the dashed line in Figure 2), the flow is predominantly vertical. Of course in the semi-circular picture of Figure 2 there is a substantial horizontal component, but this part is concentrated in the deeper part of the freatic system, where there is a preferential network and the Darcyan flow is no longer dominant. I do agree that in a situation with an impermeable foundation (without a preferential drainage structure) the flow would be partly, or even mostly, horizontal. That would be a situation in agreement with the paper of Brutsaert and Nieber (1977), which indicates a quadratic power ($n$=2 and $b$=1.5). I would expect that also in such catchments a preferential flow system is present, but because of the substantial horizontal component a quadratic power relation applies.

**6. "Residence time"**

I do agree that the term "residence time" is not correct. I shall make sure that in the final paper this term is not used in places where the time scale of the linear reservoir is meant.

Again, thank you very much for raising these very valid points and for opening the discussion, which I appreciate enormously.

**References:**

Brutsaert, W., Nieber, J. L. (1977). Regionalized drought flow hydrographs from a mature glaciated plateau. Water Resources Research, 13(3), 637-643.

Verhoest, N. E. C., and P. A. Troch (2000), Some analytical solutions of the linearized Boussinesq equation with recharge for a sloping aquifer, Water Resour. Res., 36(3), 793–800, doi:10.1029/1999WR900317.

Ye, S., Li, H. Y., Huang, M., Ali, M., Leng, G., Leung, L. R., ... Sivapalan, M. (2014). Regionalization of subsurface stormflow parameters of hydrologic models: Derivation from regional analysis of streamflow recession curves. Journal of hydrology, 519, 670-682.

---

## Author Comment (AC2) · 14 Oct 2017

I am very grateful to Dr Cuthbert for sharing his detailed reading and valuable thoughts and suggestions. Indeed, this opinion paper does not claim to have found the only explanation for linear (or non-linear) behaviour of groundwater drainage in catchments, but rather would like to present an alternative perspective and an alternative explanation for how the laboratory scale Darcy equation can be made to match with linear macro-scale behaviour. The conceptual connection, as presented in Figure 2, is the fact that in unconfined aquifers (without an impermeable base) streamlines start predominantly vertical. Also it requires an equally distributed resistance to entering the preferential drainage network. Equal resistance implies a constant proportion of $W$ (the distance to entering the preferential drainage network) and Darcy's $k$ (the conduc-

tance). As a result, *W* and *k* can both vary, but in a fixed proportion. One could imagine that parts of the catchment that are further removed from the drainage network have a larger head *H*, a larger *W* and maybe another *k*. But then the integration assuming the resistance as a constant cannot be made as simply. This is something to be further investigated. There may be more solutions of the integral of *v*d*A* that result in a linear reservoir while prescribing a certain relation between *H*, *W* and *k*. The essence, however, is that the groundwater system has structure and that approaching it purely from a Darcy perspective (the lab scale) denies the fact that all formation through which water flows have structure (except in most groundwater models).

I fully agree with the excellent summary presented by Dr Cuthbert under item 1. The Boussinesq approach still assumes that no structure is present in the groundwater body. In situations where water flows through an erodible or freely shaping material, there is always structure and there are always patterns. Most likely this has to do with systems evolving to a state of Maximum Power, as discussed and made plausible by Axel Kleidon (2016). Darcyan flow is reserved for lab experiments or for flow through a medium that has not yet evolved into a fractal-like structure due to exceedingly long morphological time scales.

I fully agree with Dr Cuthbert on item 2. The assumption is that the extremes of the groundwater system, near the phreatic table, can be described as an 'equivalent porous medium', even if there may be, and probably is, a dual porosity.

Regarding item 3, I already addressed this in my reply to Wouter Berghuijs. In addition to that, I don't think that a slope of the phreatic table implies a horizontal flow system. The flow and the streamlines are perpendicular to the equipotential lines and even in large aquifer systems, these streamlines start perpendicular to the phreatic table (unless there is substantial recharge, but this does not apply to the recession stage). It then essentially depends on the boundaries of the water body; whether there is an impermeable base, and how deep it is? Boussinesq-type solutions all assume an impermeable base, which forces streamline to be directed in the direction of the

hillslope.

I agree that there may be 'a more complex interaction between the hydraulic response in the higher and lower permeability zones', as I indicated in the beginning of this reply. I also think that the region closer to the stream may have a lower conductance, offsetting the shorter drainage length and thus maintaining a constant resistance.

Thank you for drawing the attention to the article by Swanson and Bahr (2004). This is indeed a very interesting analytical study, strengthening the notion that subsurface structures are present even in catchments with relatively small hydraulic gradients.

On item 5, I fully agree that the term recharge is not correct. I intended the flow in the upper part of the streamlines. The recharge through the unsaturated zone is not relevant to the argument. I shall adjust the text accordingly.

I also agree that the term 'residence time' is not correct and confusing. It shall be replaced. The variable $K$ is indeed the hydraulic response time of wave propagation, which is much shorter than the residence time of individual water particles.

I also agree with the well-worded summary. I would like to add, that in view of the long morphological time scales of groundwater systems, which function at geological time scales, even longer than the morphological time scales of surface drainage systems, we cannot assume that a complete subsurface drainage network is always present. If further we assume that this network makes use of cracks and fissure present in the base rock, but further, most likely, expands and develops by minerals going into solution, than these networks never stop to develop, while refining and expanding the fractal structure. In relatively young catchments such structures may not yet have been developed to the full extent. I think it is an interesting venue of research to study the expansion of such networks as a function of the mineral composition of the groundwater feeding the stream network.

Reference:

Kleidon, A. (2016). Thermodynamic foundations of the Earth system. Cambridge University Press.

Swanson, S. K., Bahr, J. M. (2004). Analytical and numerical models to explain steady rates of spring flow. Groundwater, 42(5), 747-759.

---

## Short Comment (SC4) · 20 Oct 2017

Many thanks to Prof. Savenije for his engaging reponse. Overall I think it's a nice thought experiment which I hope will open up new conceptual and numerical approaches to understanding groundwater drainage. I look forward to following the developments...
* * *

---

## Referee Comment (RC1) · S. Hergarten (Referee) · 1 Nov 2017

The opinion paper by H.H.G. Savenije addresses one of the fundamental questions in subsurface hydrology – how can we upscale equations and parameter values from the laboratory scale to larger scales? I enjoyed reading this rather short and well-written paper, although (or perhaps because) the title pulled my thoughts into a completely different direction before starting to read.

Fortunately, W.R. Berghuijs and M. Cuthbert already did an excellent job in reviewing the paper as part of the interactive discussion, so that there is not much left for me to be done. In both comments, the consideration of the linear storage as a paradigm for the system scale plays a central part, and I am also not sure whether the linear storage

is a good thematic anchor for the very interesting part presented in Sect. 3.

As pointed out by M. Cuthbert, each diffusive system of finite size turns into an exponential decay that looks like a linear storage as soon as its diffusive length scale becomes much larger than the system size. So each system of Darcy-type storages coupled by a fast preferential flow system should look like a linear storage at the long time scale where the coefficient of recession is governed by the largest Darcy-type element. I would suspect that the fast preferential flow system plays a minor part at the long time scale. This behavior was, e.g. exemplified by a fractal pattern of Darcy-type blocks where the fractal dimension of the block size distribution determines the power-law exponent of the short-term recession curve, while the size of the largest blocks is responsible for the long-term recession coefficient (Hergarten and Birk, 2007). Sorry for citing my own work here, there is probably much more literature in this direction.

Taking this into account I wonder whether the exponential decay of a linear reservoir is really our target. So far I believed that the behavior at shorter times carries much more information about the internal structure of the system. So I am aware that it is an opinion paper and not a review paper, but I am a bit afraid that evidence for any phenomenon or process-based idea could be focused too much on a single property.

For me, drawing attention on the formation of preferential flow patterns and the consequences for the resulting system characteristics is the most important contribution of this paper. But while reading this very interesting section I was a bit confused. Is the concept of a "self-organized" distribution of porosity or permeability according to minimum energy dissipation proposed by Hergarten et al. (2014) not already very close to what you are thinking of? It is, of course, only a theoretical concept; evidence is rather weak so far (Hergarten et al., 2016), and the question how to upscale it finally is still open.

But anyway, both the minimum energy expenditure of Rodriguez-Iturbe and Rinaldo and the concept of minimum energy dissipation in subsurface flow predict a strong de-

pendency of resistance on discharge. The preferred flow paths must have the lowest resistance. So the conjecture of a constant resistance made in Sect. 3 clearly contradicts to the theoretical concepts of minimum energy dissipation and perhaps even to the concept of preferential flow. As far as I can see, this also affects the arguments pointing in direction of a linear reservoir.

In summary, I still find this opinion paper a valuable contribution to the discussion, and looking at the comments by W.R. Berghuijs and M. Cuthbert I think it already has done a good job. Nevertheless, I think the points discussed above should be addressed a more thoroughly in a revised version.

References:

Hergarten, S. & S. Birk (2007). A fractal approach to the recession of spring hydrographs. Geophys. Res. Lett, 34: L11401, doi 10.1029/2007GL030097

Hergarten, S., G. Winkler & S. Birk (2014). Transferring the concept of minimum energy dissipation from river networks to subsurface flow patterns. Hydrol. Earth Syst. Sci., 18: 4277-4288, doi 10.5194/hess-18-4277-2014

Hergarten, S., G. Winkler & S. Birk (2016). Scale invariance of subsurface flow patterns and its limitation. Water Resour. Res. 52(5): 3881-3887, doi 10.1002/2015WR017530

Again sorry for citing only own work.
* * *

---

## Referee Comment (RC2) · A. Kleidon (Referee) · 10 Nov 2017

This Opinion manuscript describes an attempt to link Darcy's law that describes water flow at a small scale to the linear reservoir at the scale of a whole catchment. The main idea described here is that it is through the assumption of a constant resistance term along the flow that results in the linear reservoir, motivated by the minimum energy expenditure conjecture in works on river networks.

The manuscript is nicely and clearly written, and I found it stimulating to read. I do not really have much to say about this manuscript and recommend publication after a minor revision. I think it will be a very nice Opinion that will stimulate further research.

I think one aspect that would be nice to be added in a revision would be to describe

a bit more extensively at the end on how the critical assumption that the resistance to drainage is constant could be tested by using observations and/or models. Also, I think it would also nice if the author could add some thoughts on what the evolutionary dynamics may be that would lead to a constant resistance to drainage. These two points would help to encourage the reader how the opinion described here could be tested and extended in future work.

Minor comments:

line 36: "found" not "find" line 61: I think the correct term used by Rinaldo and Rodriguez-Iturbe is "minimum energy expenditure", not "minimum energy production". line 92: I think you intended to refer to Kleidon et al. (2013), not Kleidon and Renner (2013) line 124: A data source for the runoff series would be nice. line 131: "(is dynamic)" - do you mean "(that is, dynamic)"? line 134: I think a "tau" would be nicer than a "K", as the letter K is typically associated with a conductivity, and tau with a time scale. As you talk about a time scale, "tau" would be more appropriate. line 201: I do not know what "freatic" is - briefly explain?

---

## Author Comment (AC3) · 13 Nov 2017

I thank dr. Hergarten for his thought-provoking review. I am familiar with his work and, indeed, it would have been appropriate to refer to it. I shall do so in the final document. I do agree that the recession is dominated by the slowest part of the drainage system. Hence, in my conceptual figure 2, it is the drainage towards the preferential, fractal-like, structure that determines the time scale during recession. This is precisely what the article assumes: It is the Darcian flow in the top layers of the phreatic aquifer towards the dendritic preferential network that determines the time scale of the recession. Of course also the top layer may have some form of preferential flow, but this can be parametrized by Darcy's law and is substantially slower than the flow through the dendritic network. As Hergarten et al. (2014) mention, the origin of the dendritic network

[Figure]

**HESSD**

may be due to both predesign and solution of the substrate by the (aggressive) rainwater. I concur with Hergarten that the self-organisation of the flow pattern is most likely caused by solution of the substrate. As a result, the resistance within the drainage network is negligible compared to the resistance towards entering the network.

My article assumes that the flow towards the network is Darcian (even if partly preferential) and that it is predominantly vertical. This applies to a situation where there is no shallow impermeable boundary underlying the hillslope. If such a boundary were there, then the flow lines would be forced into a more horizontal direction, which would lead to non-linear behaviour. However, my paper is intended for catchments that do drain as a linear reservoir. Subsequently, the linear recession can be explained by assuming a phreatic aquifer draining to a dendritic subterranean network with homogeneously distributed resistance to entering this network.

Like in the papers by Hergarten et al., there is a dependence of the discharge on resistance, but the resistance $r_g$ is not the resistance within the dendritic network, but rather the resistance in the top layer of the phreatic aquifer that separates the water table from the dendritic network. We don't know how thick this layer is, but we assume that the resistance to entering the drainage network is constant in space. I don't see how this is in contradiction with the concept of minimum energy dissipation. The minimum energy dissipation applies to the dendritic network, much like the networks described by Hergarten et al. (2014), but not to the layer that separates the water table from entering the dendritic network.

Since the rate at which the dendritic network expands by solution of minerals is very slow, probably not observable at human time scales, we may assume the drainage network to be static and $r_g$ to be constant. But at geological timescales, the resistance $r_g$ is likely to reduce over time.

So in summary, I am grateful for the comments raised by Dr Hergarten and I shall incorporate his suggestions in the revised paper.

Reference:

Hergarten, S., G. Winkler S. Birk (2014). Transferring the concept of minimum energy dissipation from river networks to subsurface flow patterns. Hydrol. Earth Syst. Sci., 18: 4277-4288, doi 10.5194/hess-18-4277-2014

---

## Referee Comment (RC3) · S. Hergarten (Referee) · 19 Nov 2017

The role of the upper (Darcy-type) layer is indeed a good point. Is it really a distinct layer of constant thickness completely unaffected by the lower preferential flow network? Or is it just a "less optimized layer", in which case its properties and thickness could be spatially varying and be related to the properties of the lower region? I look forward to the discussion of such aspects in the final version of the paper.

---

## Author Comment (AC4) · 22 Jan 2018

Thank you very much for the comments and suggestions. They shall be incorporated in the revised paper.

What is the evolutionary dynamics of the drainage network?

It is likely that the drainage network makes use of cracks and fissure present in the base rock, but subsequently expands and develops by minerals going into solution. As a result, these networks never stop to develop, continuously refining and expanding the fractal structure. In relatively young catchments such structures may not yet have been developed to the full extent. By sampling the chemical contents of springs and base flow at the outfall of catchments, we may be able to determine the rate of growth of the

drainage network, and – if the mineral content of the substrate is known – the origin of the erosion material. I think it is an interesting venue of research to study the expansion of such networks as a function of the mineral composition of the groundwater feeding the stream network, possibly supported by targeted use of unique tracers.

Whether it is at all possible to test the hypothesis of constant resistance by direct observations seems doubtful. The process manifests itself at system scale and this is difficult to test by observations in the field by – for example – observations in, or samples from, piezometers. Unique tracers may provide supporting evidence, but also here, the heterogeneity is so large at small scale, that such observations, already difficult to set-up, would probably not convey much. I think that observation of the mineral composition of the drainage water, whether from springs or at the outfall, would possibly be more valuable.

---

## Author Response (AR1)

Dear editor,

In this revised paper, I have addressed most if not all the comments made during the discussion by the reviewers and by the commenters. Some of the items in the discussion belong in the discussion, and are not fit to be included in the paper, because that would shift the focus of the opinion paper. What I did, was refer the readers to the discussion for additional background on the debate.

I also tried to address the observations you made in your editor review, but not too extensively, also to prevent that the paper becomes too wide in its focus. I hope you can agree with that.

I very much appreciated the valuable discussion, both by the reviewers and the spontaneous reactions received from the audience. I appreciated the spontaneous contributions of Berghuijs and of Cuthbert enormously.

Sincerely,
Hubert Savenije